# Enzymatic Adaptation of *Bifidobacterium bifidum* to Host Glycans, Viewed from Glycoside Hydrolyases and Carbohydrate-Binding Modules

**DOI:** 10.3390/microorganisms8040481

**Published:** 2020-03-28

**Authors:** Toshihiko Katoh, Miriam N. Ojima, Mikiyasu Sakanaka, Hisashi Ashida, Aina Gotoh, Takane Katayama

**Affiliations:** 1Graduate School of Biostudies, Kyoto University, Sakyo-ku, Kyoto 606-8502, Japan; tkatoh@lif.kyoto-u.ac.jp (T.K.); ojima.nozomi.78a@st.kyoto-u.ac.jp (M.N.O.); gotoh.aina.5z@kyoto-u.ac.jp (A.G.); 2National Food Institute, Technical University of Denmark, Kemitorvet, DK-2800 Kgs. Lyngby, Denmark; miksak@dtu.dk; 3Research Institute for Bioresources and Biotechnology, Ishikawa Prefectural University, Nonoichi, Ishikawa 921-8836, Japan; 4Faculty of Biology-Oriented Science and Technology, Kindai University, Kinokawa, Wakayama 649-6493, Japan; ashida@waka.kindai.ac.jp

**Keywords:** *Bifidobacterium bifidum*, human milk oligosaccharide, mucin *O*-glycan, glycoside hydrolase, carbohydrate-binding module, cross-feeding

## Abstract

Certain species of the genus *Bifidobacterium* represent human symbionts. Many studies have shown that the establishment of symbiosis with such bifidobacterial species confers various beneficial effects on human health. Among the more than ten (sub)species of human gut-associated *Bifidobacterium* that have significantly varied genetic characteristics at the species level, *Bifidobacterium bifidum* is unique in that it is found in the intestines of a wide age group, ranging from infants to adults. This species is likely to have adapted to efficiently degrade host-derived carbohydrate chains, such as human milk oligosaccharides (HMOs) and mucin *O*-glycans, which enabled the longitudinal colonization of intestines. The ability of this species to assimilate various host glycans can be attributed to the possession of an adequate set of extracellular glycoside hydrolases (GHs). Importantly, the polypeptides of those glycosidases frequently contain carbohydrate-binding modules (CBMs) with deduced affinities to the target glycans, which is also a distinct characteristic of this species among members of human gut-associated bifidobacteria. This review firstly describes the prevalence and distribution of *B. bifidum* in the human gut and then explains the enzymatic machinery that *B. bifidum* has developed for host glycan degradation by referring to the functions of GHs and CBMs. Finally, we show the data of co-culture experiments using host-derived glycans as carbon sources, which underpin the interesting altruistic behavior of this species as a cross-feeder.

## 1. Introduction

Bifidobacteria are Gram-positive obligate anaerobes that represent common inhabitants of the gastrointestinal tract of animals [1]. In humans, it is well known that a bifidobacteria-rich microbiota (so-called, bifidus-flora) is formed in the intestinal tracts of breast-fed infants. These “first colonizers” are believed to confer various beneficial effects on human health, including protection against pathogen infection [2,3], nutritional supplementation [4], reduced inflammation [5,6], and the development of the immune system [7]. Notably, a recent research revealed that bifidus-flora formation is associated with the increased fecal concentration of indole lactic acid, a ligand of arylhydrocarbon receptor (AhR) [8]. In the *Bifidobacterium* genus, four species, i.e., *Bifidobacterium bifidum*, *Bifidobacterium breve*, *Bifidobacterium longum* ssp. *infantis* (*B. infantis*), *Bifidobacterium longum* ssp. *longum* (*B. longum*), are frequently detected in the stool of breast-fed infants. Many studies [9,10,11,12,13,14,15,16,17,18,19] have indicated that the formation of a bifidus-flora in the gut of breast-fed infants can be attributed to human milk oligosaccharides (HMOs), which are the third most abundant solid component in breastmilk after lactose and lipids [20,21,22]. Mothers produce the energy-rich HMOs, even though HMOs have no direct nutritional value for infants, as HMOs are resistant to digestive enzymes secreted in the gastrointestinal tract. Recent studies have unequivocally revealed that HMOs are utilized by infant gut-associated bifidobacteria to proliferate in their specific ecosystem [23,24]. Interestingly, the pathways for HMO assimilation differ among the *Bifidobacterium* species and even among strains of the same species, suggesting strong selective pressure and adaptive evolution under symbioses with individuals with different gut environments [11,13]. After weaning, the bifidobacterial population gradually decreases and its dominance is replaced with other microbes that are adapted to plant-derived glycan utilization, such as *Bacteroides* and *Clostridium*, to form an adult-type microbiota. Nonetheless, in many cases, *Bifidobacterium* continues to be a member of the human gut microbiome (described later).

Among *Bifidobacterium* species, *B. bifidum* is quite unique in that this species possesses many extracellular glycosidases specified for degrading host-derived glycans, including HMOs, the sugar chains of high molecular-weight glycoproteins, and glycosphingolipids [11,25,26,27]. *B. bifidum* liberates mono- and disaccharides from host glycans and releases them into the living environmental niches or culture medium in vitro. By virtue of these characteristics, *B. bifidum* can support the growth of other bifidobacteria by sharing the glycan degradation products within the community [28]. In this review, when compared with other human gut-associated *Bifidobacterium* species, we reveal the unique characteristics of *B. bifidum* that have adaptively evolved for host glycan utilization.

## 2. Prevalence and Dominance of *B. bifidum* in the Gut

The abundance of the genus *Bifidobacterium* in human fecal samples is reported to vary by country [29,30] or dwelling environment (urban industrialized area vs. remote/rural area) [31]. The factors that affect the prevalence and dominance of bifidobacteria include not only diet, birth delivery mode, and living environment but also include host genetic statuses. For example, studies have reported that secretor status or lactose intolerance (i.e., *FUT2* [32] or *LCT* [33]) is associated with the intestinal bifidobacterial population. However, it should be mentioned that the variability of the bifidobacterial abundance across the studies can be caused by varied sample processing and/or primer design in 16S metagenomic analyses [34].

*B. bifidum* is a common colonizer of the gut of mammals, including humans [35,36,37]. This species is, in general, detected in healthy individuals with a high inter-subject variability. Matsuki et al. [38] reported that *B. bifidum* was detected in 38% and 22% of adult and infant subjects, respectively. In a report examining its dominance in the intestinal samples of Finnish adults, *B. bifidum* was found to be the third most abundant, after *B. longum* and *Bifidobacterium adolescentis* [39]. Guglielmetti et al. [40] quantified the cell numbers of *Bifidobacterium* species in fecal samples from healthy male volunteers in Italy and demonstrated that *B. bifidum* was less represented than *B. longum* and *B. adolescentis* by approximately two orders of magnitude. Turroni et al. [35] have examined the diversity of bifidobacterial populations in mucosal and fecal samples of the human intestinal tract by culture-based analysis, and isolated *B. bifidum* from both mucosal and fecal samples, suggesting that this species has a mucosa-adherent property. This mucosa-adherent property is conferred by the presence of sortase-dependent pili, which may be involved in the attachment to the extracellular matrix components [41], and also by glycan-binding properties of extracellular glycoside hydrolases (described below) [42].

The age-related compositional changes of human gut bifidobacterial species have been investigated [43,44,45]. Kato et al. [44] have examined the prevalence of 11 different *Bifidobacterium* species/subspecies in the stools of 441 healthy Japanese subjects across a wide age range (0 to 104 years) by quantitative-PCR. With the exception of centenarians, *B. bifidum* was detected at all ages, but their prevalence was lower (28.3%) than that of the *B. longum* group *(B. longum* and *B. infantis*) (88.1%). A similar pattern was observed for the *Bifidobacterium catenulatum* group. Nagpal et al. [45] have examined the early-life dynamics of the bifidobacterial population in 76 full-term vaginally-born Japanese infants. They showed that *B. bifidum* became prevalent (~60%) six-months after birth and became less prevalent (~20%) after 3 years. *B. breve* was detected in 71.4% of children under 3 years old and its prevalence was consistently high in individuals younger than 10, but the number of *B. breve* cells decreased with age and was very scarce past the age of 50. The prevalence of *B. breve* also decreased during the transition from childhood to adulthood. 

Collectively, *B. breve* and *B. longum* are, in general, more abundantly detected than *B. bifidum* during the first three years of life. The prevalence of *B. breve* is, however, largely limited to infants. *B. longum* is widely distributed across age groups and frequently found in adult fecal samples. *B. bifidum* is also widely distributed across age groups, but with a lesser prevalence than the *B. longum* group. *B. adolescentis* and the *B. catenulatum* group are also abundant in adult intestinal microbiota. This transition of bifidobacterial species in human intestinal microbiota with age is primarily due to changes in diet and possibly host glycans (described below). Considering the age-related compositional changes and nutritional adaptation, *Bifidobacterium* species can be classified into at least four groups: the HMO-dedicated group (*B. infantis*), the dietary fiber-adapted group (*B. adolescentis*, *B. catenulatum* group, and other adult-type species except *B. longum*), the both HMO and dietary fiber-adapted group (*B. breve* and *B. longum*), and the both HMO and mucin-adapted group (*B. bifidum*). Thus, as the sole known mucin-degrader within the genus *Bifidobacterium*, *B. bifidum* is uniquely positioned in the context of carbohydrate utilization strategies. This may be the reason it is detected in a wide range of age groups, except for centenarians. It is known that intestinal mucin production becomes low in elderly adults [46]. Therefore, the nutritional switch, as well as host glycan availability, may have an impact on bifidobacterial prevalence and abundance.

## 3. Structures of Human Milk Oligosaccharides and Mucin O-Glycans

“HMOs” is a collective term for the mixture of more than 100 structurally different free oligosaccharides (degree of polymerization ≥ 3) produced in the mammary gland [20,47]. The synthesis of HMOs begins with the modification or the elongation of lactose (Galβ1-4Glc). Fucosylation occurs at the C2 position of Gal or the C3 position of Glc of lactose by the activity of fucosyltransferases (FUTs) to form 2′-fucosyllactose (2′-FL) and 3-fucosyllactose (3-FL), respectively. Sialylation with *N*-acetylneuraminic acid occurs at the C3 or C6 position of Gal of lactose to form 3′/6′-sialyllactose (SL). The elongation or branching of lactose is initiated with the addition of *N*-acetylglucosamine (GlcNAc) onto the C3 position of Gal of lactose to form lacto-*N*-triose II, then subsequent addition of Gal onto the C3 position or the C4 position of GlcNAc occurs to form the type-1 chain unit (Galβ1-3GlcNAc or lacto-*N*-biose I, LNB)-containing lacto-*N*-tetraose (LNT, Galβ1-3GlcNAcβ1-3Galβ1-4Glc), or the type-2 chain LacNAc unit (Galβ1-4GlcNAc)-containing lacto-*N*-*neo*tetraose (LN*n*T, Galβ1-4GlcNAcβ1-3Galβ1-4Glc). LNT and LN*n*T can then be subjected to further modification and/or elongation. GlcNAc branching also occurs at the C6 position of the Gal residue of Lac after the synthesis of lacto-*N*-triose II, and the branch is further elongated by the addition of Gal in a similar manner. When compared to the milk oligosaccharides of other mammals, HMOs are characterized by the richness of type-1 disaccharide units [20]. It should be noted that the genetic background of mothers significantly influences the structural variety of HMOs. For example, non-secretor *FUT2*-deficient mothers (*Se*-negative, 0% to 50%, depending upon ethnic background) do not produce H-antigen (Fucα1-2Gal-OR)-containing oligosaccharides such as 2′-FL or lacto-*N*-fucopentaose (LNFP) I in their milk [16], and *FUT3*-deficient mothers with the Lewis negative phenotype produce HMOs with differential amounts of Lewis antigen structures [48]. 

While HMOs are present as a free form in the aqueous phase of milk, mucin *O*-glycans are covalently attached to the polypeptide backbone through *N*-acetylgalactosaminyl Ser/Thr residues. *O*-Glycosylation on mucin glycoproteins densely occurs in regions that are rich in the amino acid residues of Pro/Thr/Ser (mucin domain) [49], and the resulting overall molecular structures of mucin glycoproteins are thought to be shaped like bottle brushes. *O*-Glycans are synthesized by sequential addition of monosaccharides by the orchestrated action of glycosyltransferases in mucin-producing cells, i.e., goblet cells, and they are classified into several groups based on their “core” types (primarily core 1, core 2, core 3, and core 4, as reviewed by [50,51,52]). Therefore, the structure of mucin *O*-glycans is highly heterogeneous and it is also dependent on which mucin molecules they are attached to [53,54]; however, they still share a structural similarity in part with HMOs. *O*-Glycans contain many type-1/2 LacNAc units as terminal and internal structures, and fucosylation/sialylation as terminal modifications (Figure 1). The structural similarities between HMOs and mucin *O*-glycans, which may reflect their overlapping function as a defense from pathogens by acting as anti-adhesive decoys [55], can also serve as prime targets for decomposition by glycan-degrading enzymes of *B. bifidum* to assimilate sugars. 

## 4. Glycoside Hydrolases Involved in the Degradation of HMOs/Mucin O-Glycans

Biochemical studies of glycoside hydrolases (GHs) from *B. bifidum* have revealed the mechanism of adaptation to HMOs and mucin *O*-glycans. The reports on a GH95 1,2-α-l-fucosidase (AfcA) from *B. bifidum* JCM 1254 by Katayama et al. [56] and a galactose operon containing a gene encoding a GH112 galacto-*N*-biose (GNB)/LNB phosphorylase from *B. longum* JCM 1217 by Kitaoka et al. [57] were the key findings that revived attention on HMOs by linking them to bifidobacterial biology, long after the first notion, proposed in 1954 by Gauhe et al. [58], that milk oligosaccharides serve as a bifidus factor. Since then, using the strain *B. bifidum* JCM 1254, extensive studies have been carried out and the gene products relevant to the HMO/mucin utilization pathways have been successfully isolated and characterized [26,27,59,60,61,62,63,64,65,66]. These efforts have illustrated how *B. bifidum* extracellularly degrades HMOs and mucin *O*-glycans into smaller sugars by the concerted action of the many cell surface-anchored GHs summarized in Figure 1. Terminal modifications such as fucosylation and sialylation can be removed by either a GH95 1,2-α-l-fucosidase (AfcA) [56,67], a GH29 1,3-1,4-α-l-fucosidase (AfcB) [62], or a GH33 sialidases (SiaBb1 [65,68] and SiaBb2 [42,60]) to expose the internal core structures. The internal type-1/2 LacNAc units are removed by GH20 lacto-*N*-biosidase (LnbB) to liberate LNB [66] and by a sequential digestion with GH2 β-galactosidase, BbgIII [63], and GH20 β-*N*-acetylhexosaminidase, BbhI, to liberate Gal and GlcNAc [63], respectively. Furthermore, *B. bifidum* possesses specific GHs for degrading various glycan epitopes that are frequently found on mucin *O*-glycans. GH89 α-*N*-acetylglucosaminidase acts on terminal α-linked GlcNAc linkages attached to gastric mucin *O*-glycans [59], while GH110 α-galactosidase acts on the blood group antigen B to liberate Gal [64]. Interestingly, *B. bifidum* does not have a gene encoding GH109 α-*N*-acetylgalactosaminidase that acts on the blood group antigen A [69]. A recent report [70] has shown that gut microbial enzymes convert A-antigens via a novel mechanism to H-antigens through a galactosamine intermediate involving a deacetylase and a GH36 galactosaminidase. Although *B. bifidum* has one GH36 gene, it likely encodes for raffinose synthase. 

Sialic acid residues are frequently modified with *O*-acetylation [71]. One of the GH33 sialidases, SiaBb1, contains an esterase-domain in its polypeptide sequence and was shown to remove the acetyl group from 9-*O*-acetylated *N*-acetylneuraminic acids [65]. Additionally, sulfated Gal and GlcNAc residues are present on intestinal mucin *O*-glycans [72]. We have recently identified a GH20 enzyme, BbhII, as a sulfoglycosidase that can remove terminal 6-sulfated GlcNAc residues from mucin [61].

Intracellular GH112 GNB/LNB phosphorylase acts on both GNB and LNB, converting them into galactose-1-phosphate and *N*-acetylhexosamine (GalNAc and GlcNAc, respectively) [57,73]. GNB is released from mucin core 1 by the action of extracellular GH101 endo-α-*N*-acetylgalactosaminidase (EngBF) [74], and LNB is formed not only from type-1 HMOs but also from mucin *O*-glycans by the action of LnbB. GNB and LNB are known to be imported by the GNB/LNB transporter, an ATP-binding cassette-type (ABC) transporter [75,76]. GH129 α-*N*-acetylgalactosaminidase (NagBb) is able to act on α-*N*-acetylgalactosaminyl Ser (Tn antigen) to liberate GalNAc and Ser [26,77]. GalNAc-Ser/Thr can be generated from core 3 or core 4 mucin *O*-glycans after complete trimming by concerted actions of GHs and peptidases. NagBb does not have a signal peptide, indicating that the hydrolysis of GalNAc-Ser/Thr takes place in the cytosol after the uptake of GalNAc-Ser/Thr into the cell by an unknown transporter.

## 5. Enzymatic Adaptation to HMO- and Mucin O-Glycan Degradation from the View of the “Spectra” of GHs and CBMs

The complete genomic sequence of *B. bifidum* was determined for the strain JCM 1255 by Hattori’s group [78] and for the strain PRL2010 by Ventura’s group [25], and, currently, a total of eleven sequences are available in public databases (NCBI genome assembly database, as of October 2019). A comparison of the annotated GH genes with those of other human-associated *Bifidobacterium* highlights its distinct mechanism that is exclusive to the assimilation of HMOs/mucin *O*-glycans. At the genus level, the total number of GHs ranges from 40 to 60 in the genomes of *Bifidobacterium*; however, the “GH spectrum”, or the distribution of GH families and the cellular localization of those GHs, is species-dependent and should reflect its carbohydrate preference and distinct assimilation mechanisms. As visualized in Figure 2, *B. bifidum* [strains PRL2010, LMG 13195 (= JCM 7004), and TMC 3115] has 14 extracellular GHs involved in HMO/mucin *O*-glycan degradation, as predicted in silico, whereas *B. infantis* ATCC 15697 (= JCM 1222) and *B. breve* UCC2003 are equipped with the intracellular counterpart enzymes (e.g., GH33 sialidase and GH95 1,2-α-l-fucosidase). Indeed, *B. infantis* ATCC 15697 (= JCM 1222) has been shown to assimilate HMOs but not mucin that was supplemented in vitro [25]. Furthermore, it is worth noting that although some GHs, including GH33 sialidases, GH2 β-galactosidase, and GH20 β-*N*-acetylhexosaminidase, are also able to act on complex *N*-glycans, *B. bifidum* does not contain *N*-glycan-specific enzymes such as GH38 α-mannosidase or GH18/GH85 endo-β-*N*-acetylglucosaminidase. Among infant-gut associated *Bifidobacterium*, *B. breve* and *B. infantis* possess such *N*-glycan-specific GHs and are reportedly able to grow on media containing *N*-glycosylated proteins [79]. *B. bifidum, B. breve,* and *B. infantis* are all adapted to HMO utilization. However, in terms of the degradation of glycan chains attached to glycoproteins, *B. breve* and *B. infantis* are specific to *N-*glycans, while *B. bifidum* prefers *O-*glycans to perhaps avoid competition with other *Bifidobacterium* species.

Interestingly, the number of carbohydrate-binding modules (CBMs) in *B. bifidum* is also higher than that of other species: 20–23 in *B. bifidum* vs. six in *B. infantis* ATCC 15697, eight in *B. longum* JCM 1217, and nine in *B. breve* UCC2003 (Figure 3). In general, the CBMs are present in tandem with the peptide chains of the GH domains and serve to enhance the catalytic activities of GHs by increasing their affinity toward the polymeric carbohydrates [80]. In particular, *B. bifidum* has 13 CBM32 domains that are specifically associated with the recognition of Gal/Lac/LacNAc and their derivatives [81], possibly to enhance catalytic activities for mucin *O*-glycans. The quantity of CBMs is outstanding within the genus *Bifidobacterium*. This could represent the evolutionary adaptation toward mucins and the basis for carbohydrate-preference of *B. bifidum*. Interestingly, Nishiyama et al. [42] found that SiaBb2 from *B. bifidum* can interact with α2,6-linked sialic acid and the blood group A trisaccharide moieties by the GH33 sialidase domain, not the CBM domain, suggesting that there are more unidentified protein sequences involved in the interaction with glycans.

These HMO/mucin-related GHs are distributed separately within the *B. bifidum* genomes, rather than being clustered [25]. However, HMO/mucin supplementation induces the expression of some of these genes [25,61,82]. Although the mechanism of the transcriptional regulation remains unclear, a conserved 24 bp sequence is found upstream of these genes, which suggests the presence of a common transcriptional regulator(s) with a shared ligand(s), presumably a sugar metabolite(s) [25].

A repertoire of the transporters may also be an important factor that dictates the carbohydrate availability for bacteria. Although the GNB/LNB [75,76], fucosyllactose [12,24,83,84], and LN*n*T transporters [85] have been identified in *Bifidobacterium*, our knowledge regarding HMO/mucin-related transporters remains limited. However, Turroni et al. [86] reported that the genome of *B. bifidum* PRL2010 contains 25 genes predicted to encode components of a transport system dedicated to carbohydrate uptake. These are classified into different groups: eight genes of ABC-type transporters, 12 genes for a component of the phosphoenolpyruvate-phosphotransferase systems, one major intrinsic protein, and four secondary carriers. The number of genes predicted to encode the proteins for carbohydrate uptake in *B. bifidum* PRL2010 is relatively small when compared with the numbers of such genes in other species of human-associated *Bifidobacterium*. For example, *B. infantis* ATCC 15697 has 68 genes predicted to encode proteins for carbohydrate uptake, including 20 of the solute-binding protein family 1 of the ABC transporter [9], while *B. bifidum* PRL2010 has only 4 of the solute-binding proteins [86], which is a stark contrast with the numbers of extracellular GHs (22 extracellular GHs in *B. bifidum* PRL2010 vs. 10 in *B. infantis* ATCC 15697). The differences in the diversity and richness of sugar transporters and extracellular GHs may thus be complementary to carbohydrate acquisition strategies within the bifidobacterial community, which simultaneously allows for the differentiation of accessible carbohydrates amongst the community members. Further studies determining the substrate specificity of transporters will reveal how these transporters serve to define the adaptation strategy of each *Bifidobacterium* species for diet-derived and/or host-derived carbohydrates.

## 6. Cross-Feeding of the Oligosaccharide Degradants among Bifidobacterial Community

Regarding the bifidobacterial population in breast-fed infant guts, *B. breve* and *B. longum* are generally more abundant than *B. infantis* and *B. bifidum*. This observation is not paralleled with their in vitro growth ability in HMO-containing media, i.e., *B. infantis* and *B. bifidum* grow avidly with HMOs as a sole carbon source and obtain high cell density, but *B. breve* and *B. longum* show very limited growth in single culture experiments [87]. This apparent inconsistency between the abundance in the intestinal tracts and abundance in vitro suggest that there is a more complicated mechanism underlying the bifidus-flora formation in infant guts.

The HMO consumption behavior of *B. bifidum* was analyzed by in vitro culture experiments followed by a high performance liquid chromatography analysis of the sugars in the corresponding spent media. The results reveal the release of monosaccharides (Fuc, Gal, and Glc) and the disaccharides (Lac and LNB) into the media [28,87]. *B. bifidum* grew well on HMOs; however, high amounts of Fuc and Gal remained unconsumed, even after prolonged culture. The leftover HMO mono- and disaccharidic breakdown products in the spent media were observed for four different strains of *B. bifidum*, and thus it appears to be a common characteristic of this species [28]. *B. infantis* is able to consume almost all HMOs by directly internalizing them with their highly abundant ABC transporters, and degrading them using cytoplasmic GHs [9,24]. Most of the *B. breve* strains catabolize limited species of HMOs, such as LNT and LN*n*T through direct uptake and intracellular degradation [85,88]. *B. longum* also shows limited HMO utilization ability, and, in most cases, it consumes LNT only. However, recent studies revealed that several strains of *B. breve* and *B. longum* have the ability to assimilate 2′-FL, 3-FL, lactodifucotetraose (LDFT), and LNFP I using specific ABC transporters and intracellular GHs [12,24,83]. Interestingly, the corresponding transporter (fucosyllactose transporter) was shown to be enriched in breast-fed infant guts [24].

Tannock et al. [89] reported that the proportion of *B. bifidum* within the total *Bifidobacterium* population was positively correlated with the elevated abundance of the genus *Bifidobacterium* within the total microbiota of breast-fed infant guts, suggesting that *B. bifidum* plays an important role for an efficient bifidus-flora formation. Given the capability of *B. bifidum* to liberate sugars into the spent media (the surrounding environments), it was suggested that cross-feeding may occur to mediate the increased abundance of total *Bifidobacterium*. Indeed, it was shown that sialic acid and other sugars liberated by *B. bifidum* can be utilized by *B. breve* in in vitro co-cultures and in vivo mouse models [68,90,91], though *B. breve* does not grow in single culture experiments in media supplemented with mucin as a sole carbon source. Additionally, a correlation with the higher abundance of *Bifidobacterium* in the total microbiota was observed not only for *B. bifidum* in breast-fed infants, but also for *B. longum*, *B. adolescentis*, and *B. catenulatum* group in human subjects with a wide age range [44]. Interestingly, some *B. longum* and *B. adolescentis* strains were shown to have the capability to serve as cross-feeders of plant-derived polysaccharide degradants [92,93,94]. These observations suggested that intragenus cross-feeding might continue in the bifidobacterial community after weaning. As for extracellular GHs targeting plant-derived polysaccharides, *B. longum* JCM 1217 possesses eleven extracellular GH43 β-xylosidase/α-l-arabinofuranosidases (Figure 2). *B. adolescentis* ATCC 15703 and *B. catenulatum* DSM 16992 (= JCM 1194) have two extracellular GH13 (amylase family) and one extracellular GH121 (β-l-arabinobiosidase), respectively.

A recent study suggested that cross-feeding of the HMO/mucin degradation products is not limited to the bifidobacterial community. A *Cutibacterium avidum* isolate from infant feces was shown to produce a higher amount of propionate when co-cultured with *B. bifidum* in yeast extract, casitone and fatty acid (YCFA) medium containing HMOs than when cultured alone in the same medium, suggesting inter-genus utilization of *B. bifidum*-released sugar metabolites [95].

## 7. Co-Culture Experiments

The co-culture experiments in our previous study showed that *B. bifidum* supported the growth of other bifidobacterial species. *B. longum* 105-A strain grew in medium containing 1% HMOs as the sole carbon source, but only when it is co-cultured with *B. bifidum* [28] (Figure 4A,B). The data were reproduced from [28]. The supportive growth effect of *B. bifidum* for *B. longum* was also observed in medium containing 1% porcine gastric mucin (PGM) as a sole carbon source (Figure 4C,D). To examine how LNB and GNB disaccharides liberated from HMOs and mucin *O*-glycans affect the growth of *B. longum* by cross-feeding from *B. bifidum*, we disrupted the *gltA* gene that encodes the solute-binding protein of the GNB/LNB transporter of *B. longum* [75,76] and made the knockout strain compete against the wild-type strain. Unexpectedly, in the growth competition assays in the presence of *B. bifidum*, the colony-forming units of Δ*gltA* strain at 12 h culture were comparable with that of the wild-type in the media containing either HMOs or mucin (Figure 4E,F). These results indicate that the utilization of GNB/LNB was not prioritized by *B. longum* in the presence of many other sugars, such as Gal, GlcNAc, or lactose released from HMOs/mucin *O*-glycans under our experimental conditions. It should be noted that the growth ability on LNB was completely abrogated by the disruption of *gltA* (Appendix A), which is well-conserved among the infant gut-associated *Bifidobacterium* [11,96].

Further evidence for cross-feeding between *B. bifidum* and other species was provided by fecal culture experiments [28]. We collected fecal samples from healthy Japanese infants, children, and adults, and then incubated the samples in media supplemented with Glc or HMOs as a sole carbon source, with and without the addition of *B. bifidum* strains. After 24 h of culture, we quantified the abundance of *Bifidobacterium* species other than *B. bifidum* by quantitative-polymerase chain reaction (qPCR). The results indicate that the addition of *B. bifidum* markedly increased both the abundance and ratio of endogenous *Bifidobacterium* species other than *B. bifidum* in the samples. Importantly, this enhancement was observed only for HMO-containing media but not for Glc-containing media. Additionally, in the presence of deoxyfuconojirimycin (DFJ), a potent inhibitor for α-fucosidases (GH29 1,3-1,4-α-l-fucosidase and GH95 1,2-α-l-fucosidase), the bifidus-flora formation-promoting effect exerted by *B. bifidum* was abrogated. Since most of the abundant HMOs, such as 2′-FL, 3-FL, LDFT, LNFP, and LNDFH are modified with fucose, the removal of the fucose residues should be a critical step for other GHs to further degrade the sugars. Taken together, *B. bifidum* can enrich the bifidobacterial population and abundance by acting as a cross-feeder of the HMO- and mucin *O*-glycan degradants produced by its membrane-associated GHs with CBMs.

## 8. Conclusions

*B. bifidum* resides in the gastrointestinal tract of humans of a wide age range, yet generally does not dominate the intestinal microbiota, like *B. longum* and *B. breve* do during infancy. Instead of being the dominant species in the gut ecosystem, however, it influences the microbiota composition through the altruistic cross-feeding of host glycan degradants. Apart from the microbe to microbe interactions, recent studies have revealed that *B. bifidum* has immunomodulatory activity by inducing regulatory T-cells by microbe-host interaction with its cell-surface polysaccharides [98]. The production of indole lactic acid, an AhR ligand, by infant gut-associated *Bifidobacterium* species, including *B. bifidum*, was also recently reported [8,99]. Further investigation is needed to understand the physiology of this species in the gut ecosystem and its role in host interactions.

## 9. Materials and Methods 

### 9.1. In Silico Analysis of Bifidobacterial Genes

The protein information of GHs and CBMs in *B. bifidum* PRL 2010, JCM 7004 (=LMG 13195), TMC 3115, *B. breve* UCC2003, *B. infantis* ATCC 15697 (=JCM 1222), and *B. longum* JCM 1217 were retrieved from the carbohydrate-active enzymes (CAZy) database (http://www.cazy.org) [100]. The pseudogenes that encode incomplete protein sequences (fragment) are excluded from the analysis. Protein localization was predicted by signalP-5.0 (http://www.cbs.dtu.dk/services/SignalP/) and TMHMM Server v. 2.0 (http://www.cbs.dtu.dk/services/TMHMM/). 

### 9.2. Disruption of gltA in B. longum 105-A

A markerless gene disruption of *gltA* (BL105A_1604) was carried out by the methods described previously [24]. The upstream and downstream regions of *gltA* were amplified by PCR from the genome of *B. longum* 105-A (JCM 31944) [101] using the primer pairs of Pr-15/16 (5′-cggtacccggggatcatctctactccttcgtagtgaaatc-3′ and 5′-caggcatgcaagctttaacatgcggtgtccccgttg-3′) and Pr-11/12 (5′-tatatatgagtactgatgcgaccacgcccggaatg-3′ and 5′-cgagtcgctcgaattcattaacggttagggttccttccag-3′), respectively. The underlined sequences represent 15 bp-extensions required for In-Fusion cloning (Clontech Laboratories, Mountain View, CA, USA). The amplified fragments were ligated with the replicon of the conditional replication plasmid pBS423 Δ*repA* so that the marker gene (spectinomycin resistance) was sandwiched between the upstream and downstream regions of *gltA* [24,102]. Subsequent integration into the genome and excision of the marker gene from the genome were conducted as described previously [24]. Disruption at the targeted site was confirmed by genomic PCR using the primer pair Pr-36/37 (5′-gtcgccgaagaagttcaccttg-3′ and 5′-ttctggaacctgtcttcttgattgc-3′) and Western Blot analysis using anti-GltA antibodies [87] (Appendix A). 

### 9.3. Co-Culture of B. longum with B. bifidum

The wild-type (WT) strain of *B. longum* 105-A with a plasmid carrying the spectinomycin resistance gene on a plasmid (pBFO2) [97] was cultured in basal medium [87] supplemented with 1% porcine gastric mucin (PGM, Sigma-Aldrich, MO, USA) as a carbon source in the presence and absence of *B. bifidum* JCM 1254. During incubation, aliquots were collected, serially diluted, and spread on GAM agar plates (Nissui Pharmaceutical, Co., Ltd., Tokyo, Japan) supplemented with and without 30 μg/mL of spectinomycin. The colonies appearing on the antibiotic-containing plates were attributed to *B. longum* cells, while those formed on antibiotic-free plates were assumed to represent the sum of *B. longum* and *B. bifidum* cells. When the Δ*gltA* derivative of *B. longum* 105-A was competed against its parental WT strain (with pBFO2 carrying the spectinomycin resistance gene [97]) for the growth on PGM in the presence of *B. bifidum*, pBFS38 with the chloramphenicol resistance gene [97] was used as the marker of the Δ*gltA* strain. WT and Δ*gltA* derivative were also cultured for competition in basal medium supplemented with 1% HMOs as a carbon source. During incubation, aliquots were collected, serially diluted, and spread on GAM agar plates containing the respective antibiotics to determine the CFUs of each strain. Spectinomycin and chloramphenicol were used at the final concentrations of 30 and 2.5 μg/mL, respectively.

## Figures and Tables

**Figure 1 microorganisms-08-00481-f001:**
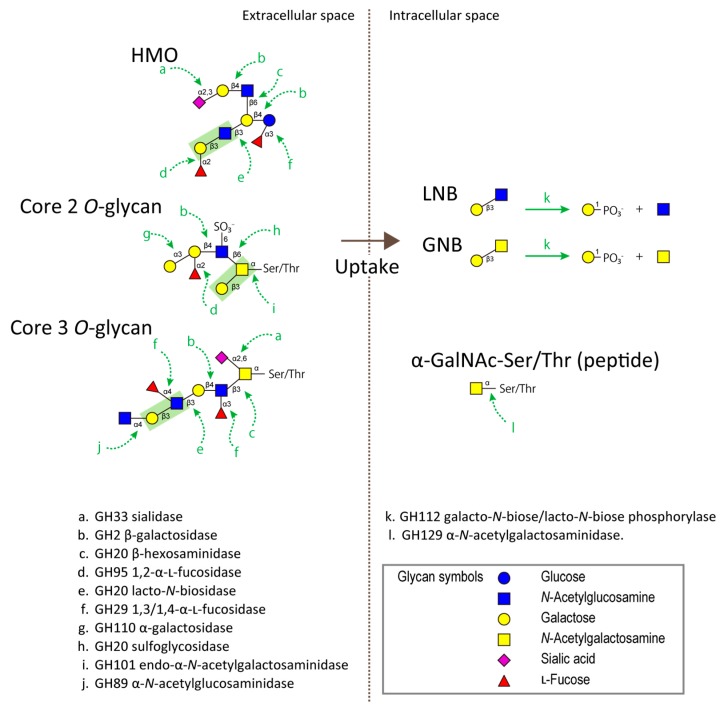
Enzymes involved in the degradation and assimilation of human milk oligosaccharides (HMOs) and mucin *O*-glycans by *B. bifidum*. The sugars are depicted according to the nomenclature committee of the Consortium for Functional Glycomics (http://www.functionalglycomics.org/static/index.shtml).

**Figure 2 microorganisms-08-00481-f002:**
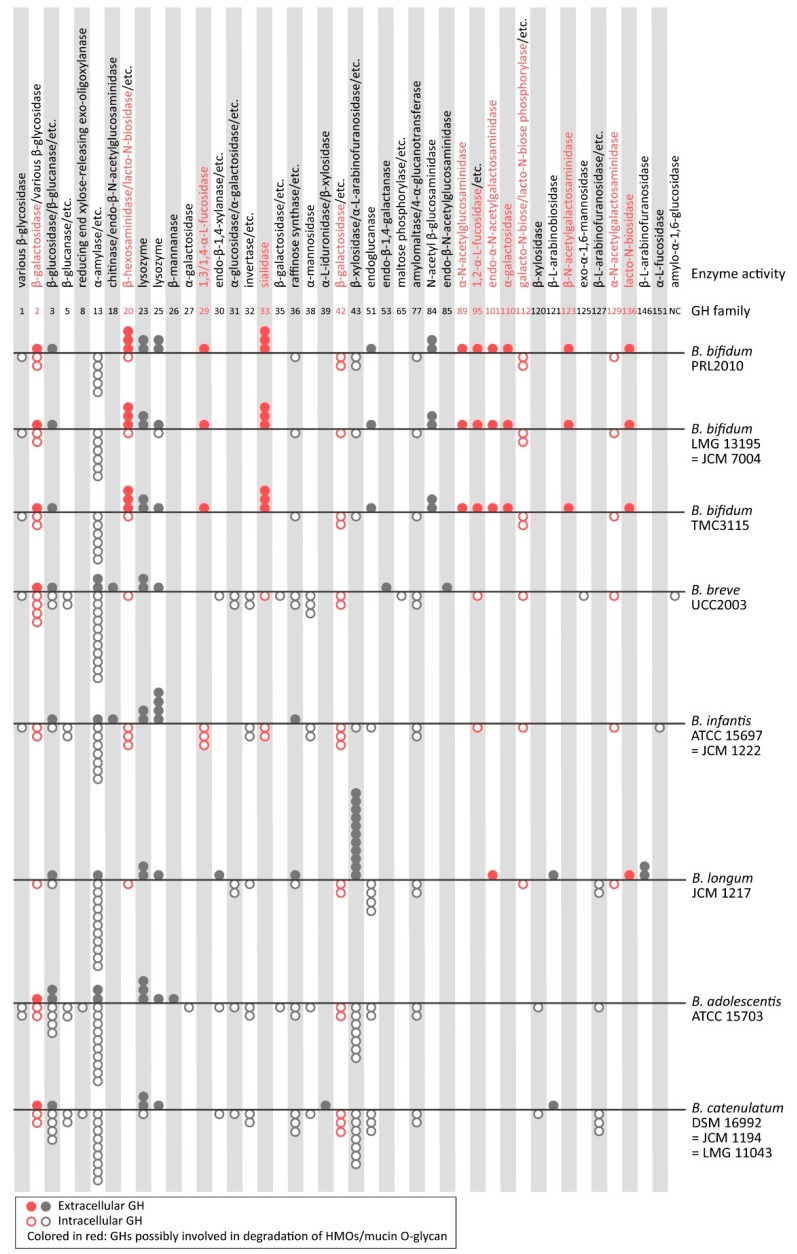
Distribution of glycoside hydrolases (GHs) and their cellular localization in the selected *Bifidobacterium* species/strains. The circles represent the occurrence of enzymes (domains) classified in the respective GH families in the CAZy database (http://www.cazy.org). The extracellular and intracellular enzymes are shown by solid and open circles, respectively. Protein localization was predicted by signalP-5.0 (http://www.cbs.dtu.dk/services/SignalP/) and TMHMM Server v. 2.0 (http://www.cbs.dtu.dk/services/TMHMM/). The pseudogenes that encode incomplete protein sequences are excluded from the analysis. The GHs that are (possibly) involved in the degradation of HMOs and mucin *O*-glycans are colored in red.

**Figure 3 microorganisms-08-00481-f003:**
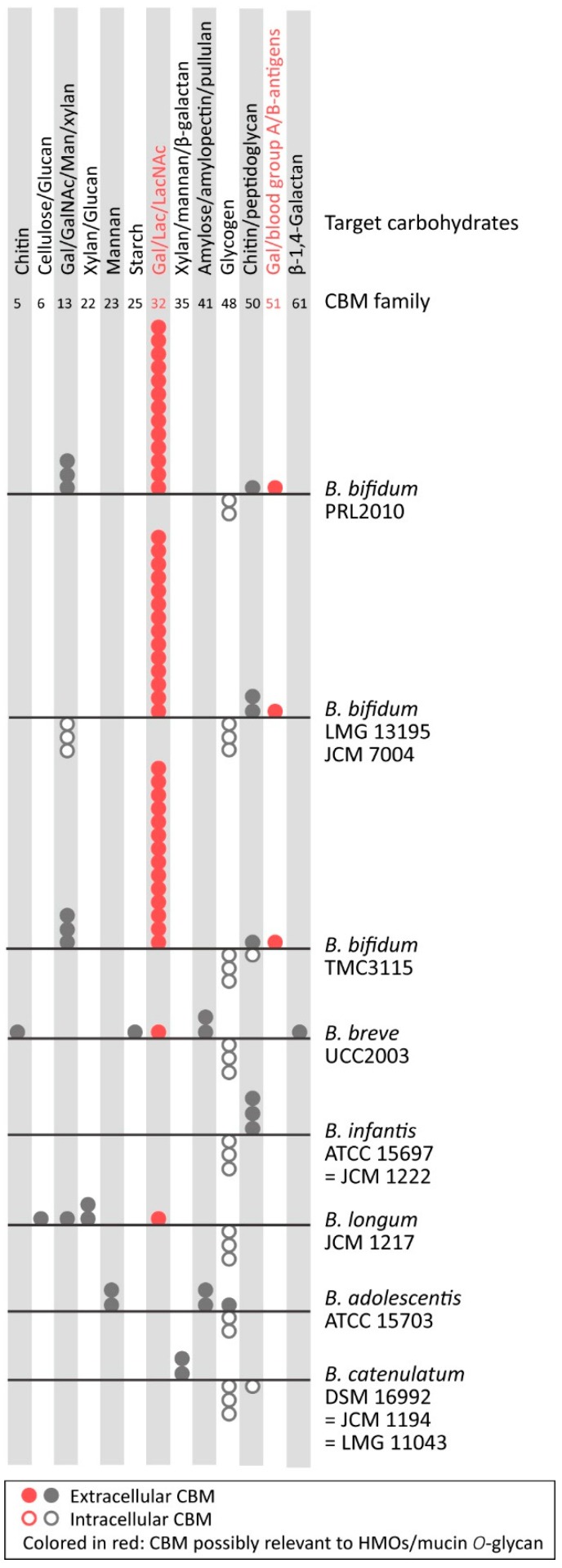
Distribution of carbohydrate-binding modules (CBMs) and their cellular localization in the selected *Bifidobacterium* species/strains. The circles represent the occurrence of CBMs (domains) classified in the CAZy database (http://www.cazy.org). The extracellular and intracellular CBMs are shown by solid and open circles, respectively. The localization of proteins including the CBMs was predicted by signalP-5.0 (http://www.cbs.dtu.dk/services/SignalP/) and TMHMM Server v. 2.0 (http://www.cbs.dtu.dk/services/TMHMM/). The pseudogenes that encode incomplete protein sequences are excluded from the analysis. The CBMs possibly associated with HMO- and mucin *O*-glycan degradation are colored in red.

**Figure 4 microorganisms-08-00481-f004:**
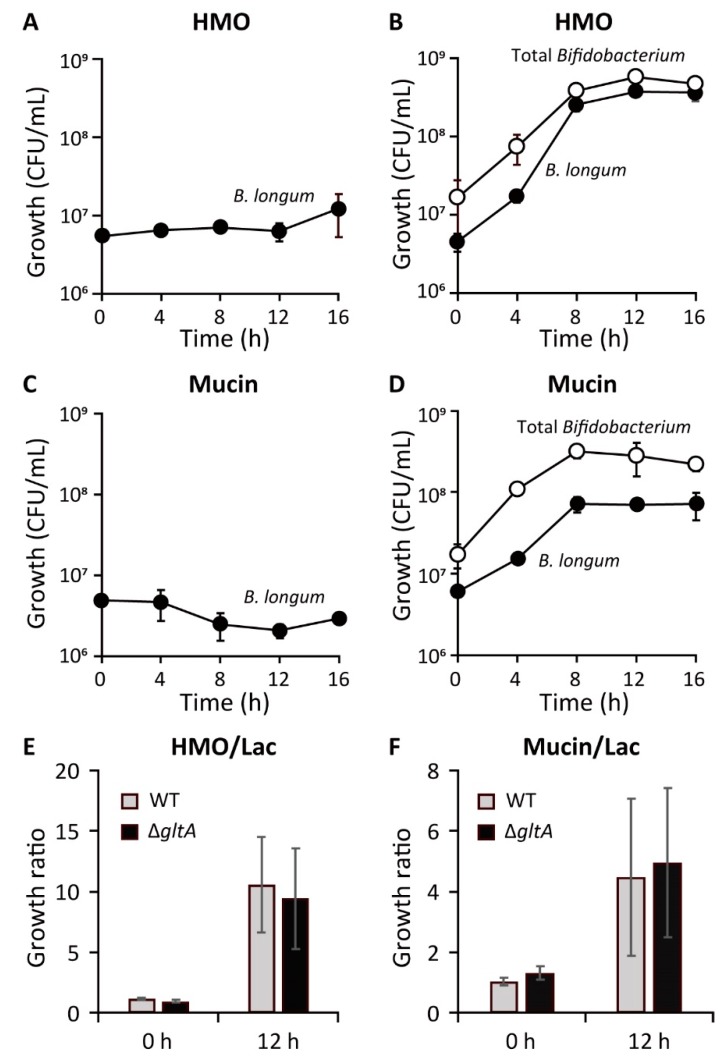
*B. bifidum*-mediated cross-feeding supports the growth of *B. longum* on HMO- and mucin-containing media. (**A**–**D**): The growth of *B. longum* 105-A in a basal medium supplemented with 1% human milk oligosaccharides (HMOs) (**A**,**B**) or 1% porcine gastric mucin (**C**,**D**) in the absence (**A**,**C**) and presence (**B**,**D**) of *B. bifidum* JCM 1254. The wild-type *B. longum* 105-A strain carrying the chloramphenicol (**A**,**B**) or spectinomycin (**C**–**F**) resistance gene on a plasmid (pBFS38 or pBFO2, respectively) [97] was used for monitoring the growth. Colony-forming units (CFU) of *B. longum* 105-A were determined by spreading the serial dilution of the cultures on the agar plates containing the antibiotics (closed circles), while the CFU of total *Bifidobacterium* was determined using the agar plates without antibiotics (open circles). The data used in (**A**) was obtained from our previous study [28]. (**E**,**F**): The growth competition between wild-type and Δ*gltA* mutant strains of *B. longum* 105-A in the presence of *B. bifidum* JCM 1254. Lactose (Lac), HMOs, or PGM was used as the sole carbon source. The ratio of the growth on HMOs (**E**) or PGM (**F**) that was normalized by that on Lac was compared. The Δ*gltA* mutant was transformed with pBFS38 [97] carrying the chloramphenicol resistance gene, and CFU was determined on agar plates containing the antibiotics.

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
