# Peer review of "Enzymatic Adaptation of Bifidobacterium bifidum to Host Glycans, Viewed from Glycoside Hydrolyases and Carbohydrate-Binding Modules"

_microorganisms, 2020, doi:10.3390/microorganisms8040481_

Round 1

Reviewer 1 Report

Dr Katoh and colleagues describe their results of co-culturing experiments of Bifidobacteria and demonstrate the relevance of human milk oligosaccharides for facilitating co-growth and enrichment of the bifidobacterial landscape. They underline the importance of metabolic cross-communication between different strains. 

The article consists of a detailed section on the biology of oligosaccharides and the various enzymatic properties underlying bifidobacterial strains followed by a description of the group`s earlier own work in this area.

The article is well written and referenced. 

Formal aspects:

Suggestions: L41 : ` in intestinal tracts of breastfed infants `

legend to figure 4: suggest using either capital or lower case letters when referring to subsections 

Author Response

Answers to the comments raised by Reviewer 1.

>Suggestions: L41 : ` in intestinal tracts of breastfed infants `

Both reviewers suggested modification of this sentence. We have followed your suggestion.

>legend to figure 4: suggest using either capital or lower case letters when referring to subsections

We have used capital letters.

Reviewer 2 Report

Enzymatic adaptation of Bifidobacterium bifidum to host glycans, viewed from glycoside hydrolyases and carbohydrate-binding modules, by Katoh et al.

This is a nice and comprehensive overview on the carbohydrate-degrading capacity of members of the Bifidobacterium bifidum species. The literature cited is appropriate and the writing is in general easy to follow and relevant. Other than a range of editorial suggestions and a couple of scientific adjustments, which are intended to improve the completeness and English narrative of the story, I do not have any major issues with this paper.

Line 18; Rather than saying that the genus Bifidobacterium is a symbiont of humans, it is better to say that certain species of the genus Bifidobacterium represent human symbionts, Similarly in the next sentence (specifically the part online 19), the authors may replace ‘Bifidobacterium’ with such bifidobacterial species’

Line 22; replace ‘This species is’ with ‘These species are’

Line 23; remove the word ‘evolutionarily’

Line 24; replace ‘could enable longitudinal colonization in’ with ‘enabled longitudinal colonization of’

Line 25; change sentence into ‘The ability of this species to assimilate various host glycans can be etc’

Line 29/30; remove ‘the’ from ‘the members’, and change ‘gut-associated Bifidobacterium’ with ‘gut-associated bifidobacteria’.

Line 31; change ‘in human guts’ to ‘in the human gut’

Line 32; delete ‘therof’

Line 39; ‘that generally inhabit gastrointestinal tracts of animals’ should be ‘that represent common inhabitants of the gastrointestinal tract of animals’

Line 40/41; should be ‘….a bifidobacteria-rich gut microbiota (so-called….) is formed in breast-fed infants’

Line 48; ‘stools’ should be ‘stool’

Line 51; replace ‘…produce HMOs by consuming a quantity of ATP,…’ by ‘…produce the energy-rich HMOs,…’, and also insert ‘direct’ between the words ‘no nutritional’.

Line 52; ‘tracts’ should be ‘tract’

Line 53; ‘equivocally’ should be ‘unequivocally’, and ‘the infant gut-associated Bifidobacterium to proliferate in infant gut ecosystems’ should be ‘infant gut-associated bifidobacterial proliferate in their specific ecosystem’

Line 61; ‘they possess’ should be ‘this species possesses

Line 66; ‘Bifidobacterium’ should be ‘bifidobacteria’, and ‘degradants’ by ‘degradation products’

Line 66/67; ‘by comparing with’ should be ‘when compared with’

Line 69; replace ‘guts’ by ‘gut’

Line 70/71; one of the main other reasons why bifidobacterial abundance varies so much is because of sample processing and primers used for 16S RNA-based amplification. There is definitely one paper that has been published on this, I think by the group of Marco Ventura.

Line 76; ‘frequent colonizer in’ should be ‘common colonizer of’

Line 77delete ‘groups of’

Line 92; ‘over a wide’ should be ‘across a wide’

Line 95; remove ‘also’

Line 108; ‘change’ should be ‘changes’

Line 109-113; the division of human-associated bifidobacterial species in the suggested groups is somewhat subjective. For example, B. breve is known to utilize various plant-derived dietary carbohydrates, such as starch, galactan, cellobiose, and stachyose/raffinose, and has rather limited abilities in terms of HMO-degradation (LNnT and sometimes 2/3-FL). This species could then be better considered to be member of the group that also includes B. longum subsp. longum.

Line 117; ‘may impact bifidobacterial composition’ should be ‘may impact on bifidobacterial prevalence and abundance’

Line 120; ‘in mammary glands’ should be ‘in the mammary gland’

Line 244; remove ‘a’ from ‘a sequential’

Line 169; ‘by a concerted’ should be ‘by the concerted’

Line 211; ‘HMOs/mucin’ should be ‘HMO/mucin’

Line 222; replace ‘prefer O-glycans to avoid’ by ’prefers O-glycans to perhaps avoid’

Line 224; delete ‘the’ in ‘of the’

Line 228; delete ‘of’

Line 254; replace ‘good contrast in’ with ‘stark contrast with’

Page 7 and 8, Figures 2 m& 3; the resolution of this figure is rather low

Line 288; insert ‘corresponding’ between the words ‘the spent’

Line 291; replace ‘HMO degradants’ by HMO monosaccharidic breakdown products’

Line 319; replace ‘degradants’ with degradation products’

Line 372; replace ‘resides in human guts across age groups, but it’ with ‘resides in the gastrointestinal tracts of humans of a wide age range, yet’

Author Response

Answers to the comments raised by Reviewer 2.

We apologize that our original manuscript contained a lot of grammatical errors, and appreciate your thorough reading and helpful comments/suggestions.

>Line 18; Rather than saying that the genus Bifidobacterium is a symbiont of humans, it is better to say that certain species of the genus Bifidobacterium represent human symbionts, Similarly in the next sentence (specifically the part online 19), the authors may replace ‘Bifidobacterium’ with such bifidobacterial species’

Thank you for the comment. In accordance with your suggestion, we have modified the sentences.

>Line 22; replace ‘This species is’ with ‘These species are’

We would like to leave it as it is, because in the preceding sentence, we have mentioned the species name of B. bifidum to focus on. In the modified sentence, ‘adapted to assimilate’ has been replaced with ‘adapted to efficiently degrade’ to represent the specific characteristic of B. bifidum.

>Line 23; remove the word ‘evolutionarily’

We have removed it.

>Line 24; replace ‘could enable longitudinal colonization in’ with ‘enabled longitudinal colonization of’

We have modified that part.

>Line 25; change sentence into ‘The ability of this species to assimilate various host glycans can be etc’

We have modified that part.

>Line 29/30; remove ‘the’ from ‘the members’, and change ‘gut-associated Bifidobacterium’ with ‘gut-associated bifidobacteria’.

We have modified that part.

>Line 31; change ‘in human guts’ to ‘in the human gut’

We have modified that part.

>Line 32; delete ‘therof’

We have deleted it.

>Line 39; ‘that generally inhabit gastrointestinal tracts of animals’ should be ‘that represent common inhabitants of the gastrointestinal tract of animals’

We have modified that part.

>Line 40/41; should be ‘….a bifidobacteria-rich gut microbiota (so-called….) is formed in breast-fed infants’

Both reviewers suggested modification of this sentence. Here we have followed the reviewer 1’s suggestion.

>Line 48; ‘stools’ should be ‘stool’

We have corrected it.

>Line 51; replace ‘…produce HMOs by consuming a quantity of ATP,…’ by ‘…produce the energy-rich HMOs,…’, and also insert ‘direct’ between the words ‘no nutritional’.

We have modified that part (Lines 51-52).

>Line 52; ‘tracts’ should be ‘tract’

We have corrected it (Line 53).

>Line 53; ‘equivocally’ should be ‘unequivocally’, and ‘the infant gut-associated Bifidobacterium to proliferate in infant gut ecosystems’ should be ‘infant gut-associated bifidobacterial proliferate in their specific ecosystem’

We have corrected them accordingly (Lines 53-54).

>Line 61; ‘they possess’ should be ‘this species possesses

We have corrected it (Line 62).

>Line 66; ‘Bifidobacterium’ should be ‘bifidobacteria’, and ‘degradants’ by ‘degradation products’

We have modified that part (Line 67).

>Line 66/67; ‘by comparing with’ should be ‘when compared with’

We have modified that part (Line 68).

>Line 69; replace ‘guts’ by ‘gut’

We have corrected it (Line 70).

>Line 70/71; one of the main other reasons why bifidobacterial abundance varies so much is because of sample processing and primers used for 16S RNA-based amplification. There is definitely one paper that has been published on this, I think by the group of Marco Ventura.

Thank you for the comment. In accordance with your suggestion, we have made a description of it and cited the relevant paper. We hope the modification is satisfactory (Lines 71 and 76-78).

>Line 76; ‘frequent colonizer in’ should be ‘common colonizer of’

We have modified that part (Line 79).

>Line 77delete ‘groups of’

We have deleted it (Line 80).

>Line 92; ‘over a wide’ should be ‘across a wide’

We have corrected it.

>Line 95; remove ‘also’

We have removed it (Line 98).

>Line 108; ‘change’ should be ‘changes’

We have corrected it (Line 111).

>Line 109-113; the division of human-associated bifidobacterial species in the suggested groups is somewhat subjective. For example, B. breve is known to utilize various plant-derived dietary carbohydrates, such as starch, galactan, cellobiose, and stachyose/raffinose, and has rather limited abilities in terms of HMO-degradation (LNnT and sometimes 2/3-FL). This species could then be better considered to be member of the group that also includes B. longum subsp. longum.

Thank you for the critical comment. We have modified that part appropriately.

>Line 117; ‘may impact bifidobacterial composition’ should be ‘may impact on bifidobacterial prevalence and abundance’

We have modified that part (Line 120-121).

>Line 120; ‘in mammary glands’ should be ‘in the mammary gland’

We have corrected it (Line 124).

>Line 144; remove ‘a’ from ‘a sequential’

We have removed it (Line 148)..

>Line 169; ‘by a concerted’ should be ‘by the concerted’

We have corrected it (Line 173).

>Line 211; ‘HMOs/mucin’ should be ‘HMO/mucin’

We have corrected it (Line215).

>Line 222; replace ‘prefer O-glycans to avoid’ by ’prefers O-glycans to perhaps avoid’

We have modified that part (Line 226-227).

>Line 224; delete ‘the’ in ‘of the’

We have deleted it (Line 228).

>Line 228; delete ‘of’

We have deleted it (Line 232).

>Line 254; replace ‘good contrast in’ with ‘stark contrast with’

We have modified that part (Line 258).

>Page 7 and 8, Figures 2 m& 3; the resolution of this figure is rather low

The figures have been created using Adobe Illustrator, which we have uploaded onto the journal website.

>Line 288; insert ‘corresponding’ between the words ‘the spent’

We have inserted the word (Line 292).

>Line 291; replace ‘HMO degradants’ by HMO monosaccharidic breakdown products’

We have modified that part in accordance with your suggestion (Line 295).

>Line 319; replace ‘degradants’ with degradation products’

We have modified that part (Line 324).

>Line 372; replace ‘resides in human guts across age groups, but it’ with ‘resides in the gastrointestinal tracts of humans of a wide age range, yet’

We have modified that part (Line 362).